# Use of Time-of-Flight Ultrasound to Measure Wave Speed in Poplar Seedlings

**Fenglu Liu [1,2], Pengfei Xu [1,2], Houjiang Zhang [1,2,\*], Cheng Guan [1,2], Dan Feng [1,2] and Xiping Wang [3]**

1 School of Technology, Beijing Forestry University, Beijing 100083, China
2 Joint International Research Institute of Wood Nondestructive Testing and Evaluation, Beijing Forestry University, Beijing 100083, China
3 USDA Forest Service, Forest Products Laboratory, Madison, WI 53726-2398, USA
* Correspondence: hjzhang6@bjfu.edu.cn; Tel.: +86-10-6233-6925

**Abstract:** In this study, 145 poplar (*Populus × euramericana cv.*'74/76') seedlings, a common plantation tree species in China, were selected and their ultrasonic velocities were measured at four timepoints during the first growth year. After that, 60 poplar seedlings were randomly selected and cut down to determine their acoustic velocity, using the acoustic resonance method. The effects of influencing factors such as wood green density, microfibril angle, growth days, and root-collar diameter on acoustic speed in seedlings and the relationship between ultrasonic speed and acoustic resonance speed were investigated and analyzed in this work. The number of specimens used for investigating growth days and root-collar diameter was 145 in both cases, while 60 and two specimens were used for investigating wood density and the microfibril angle, respectively. The results of this study showed that the ultrasonic speed of poplar seedlings significantly and linearly increased with growth days, within 209 growing days. The ultrasonic velocity of poplar seedlings has a high and positive correlation with growth days, and the correlation was 0.99. However, no significant relationship was found between the ultrasonic velocity and root-collar diameter of poplar seedlings. Furthermore, a low and negative relationship was found between wood density and ultrasonic speed ($R^2 = 0.26$). However, ultrasonic velocity significantly decreased with increasing microfibril angle (MFA) in two seedlings, and thus MFA may have an impact on ultrasonic speed in poplar seedlings. In addition, ultrasonic velocity was found to have a strong correlation with acoustic resonance velocity ($R^2 = 0.81$) and a good correlation, $R^2 = 0.75$, was also found between the dynamic moduli of elasticity from ultrasonic and acoustic resonance tests. The results of this study indicate that the ultrasonic technique can possibly be used to measure the ultrasound speed of young seedlings, and thus early screen seedlings for their stiffness properties in the future.

**Keywords:** ultrasonic speed; poplar seedlings; acoustic resonance; density; microfibril angle; root-collar diameter

## 1. Introduction

Trees grown in plantation forests are expected to have a high average value of stiffness (i.e., a high modulus of elasticity (MOE)), a low microfibril angle (MFA), and a low shrinkage propensity to distort, thus yielding lumber or other wood products with a high quality, as well as resulting in a high average grade out-turn [1]. The genetics have a significant impact on whether the tree can produce an acceptable yield of wood products, such as lumbers. Therefore, it is extremely desirable to be able to select young seedlings that are more likely to produce better wood products and, hence, can be grown to maturity in the knowledge that they will be more valuable [1]. Many studies show that for

juvenile wood, MFA decreases with age [2,3]. There are many studies showing that it is possible to enable an early selection of trees that yield better wood quality. For example, Donaldson and Burdon found that it may be possible to effectively select for desirable MFA, starting from ring 1 [4]. Similarly, Dungey et al. found that it was possible to early select for high stiffness (MOE) in *Pinus radiata* D. Don, around rings 4 to 8 at breast height [5]. In addition, Nakada studied *Cryptomeria japonica* D. Don and found an effective selection method and improvement in average log stiffness at an early age (5 years old). He also suggested that it was worth developing an early selection for tree quality [6]. Moreover, Watt et al. (2010) suggested that *P. radiata* clones with high average stiffness can be selected from trees aged 5 years [7]. Emms et al. stated that the early screening of tree quality has benefits to the forestry industry, potentially yielding better wood quality [1,8].

Acoustic techniques such as longitudinal stress waves, acoustic resonance, and ultrasound have been used to determine the mechanical properties of wood materials for many years [9,10]. Since the acoustic speed of wood materials is related to their stiffness, the measurement of acoustic speed in wood still receives much attention from researchers [11–13]. For instance, the acoustic speeds measured by longitudinal stress waves, acoustic resonance or ultrasound are widely used to evaluate the quality of standing trees and to assess the mechanical performance of logs and lumbers and further sort their grades in terms of measured MOE [14–19]. Therefore, acoustic technologies including longitudinal stress waves, acoustic resonance, and ultrasound have been well developed for standing trees, logs, lumbers, and other wood products. For seedlings, however, few acoustic techniques are applied to evaluate their wood quality. There is a lack of interest in the mechanical properties of seedlings and methods for quality evaluation; hence, there are few studies on acoustic speed measurement and wood quality assessment for seedlings. Only Emms et al. measured the acoustic speeds in 2-year-old *Pinus radiata* seedlings, using a longitudinal-wave time-of-flight prototype that they built and an acoustic resonance technique [1,8]. They found that this longitudinal-wave time-of-flight acoustic technique may be able to become a novel technique for non-damaging measurement of acoustic speed in seedlings, and that the technique shows good promise as a rapid and cost-effective tool for early screening of wood quality. Furthermore, they suggest that the measurement of acoustic speed in seedlings has benefits to the forestry industry, potentially enabling the early selection of trees that yield better quality wood. Huang et al. used the stress-wave technique to measure the stiffness properties of seedlings, and Divos et al. conducted seedling segregation by acoustic velocity using stress-wave devices [20,21]. Therefore, more efforts are still needed to find a proper acoustic technique for non-destructive measurement of sound speed, and to establish a comprehensive evaluation method for the quality of seedlings.

It is well-known that acoustic speed is related to the important quality properties of wood, such as stiffness, referred to as MOE, grain angle and the MFA of the S2 layer in the cell wall. Therefore, it is necessary to accurately measure acoustic speed in seedlings and to investigate the influencing factors on acoustic speed, since this will help to improve the reliability and accuracy of assessment for the quality of seedlings via acoustic techniques. However, there are few reports about acoustic speed measurement in seedlings using acoustic technology, especially for ultrasound, and no reports were found for studying the influencing factors on acoustic speed in seedlings. As mentioned previously, only Emms et al. conducted acoustic speed measurement on *Pinus radiata* seedlings, using longitudinal waves and an acoustic resonance technique [1,8]. Huang et al. used the stress-wave method to test the mechanical properties of seedlings, and Divos et al. performed seedling segregation according to stress-wave acoustic velocity [20,21]. Therefore, to the best of our knowledge, no research on acoustic speed measurement for seedlings using ultrasound was found, and no paper investigating influencing factors, such as wood density, MFA, growth days, and root-collar diameter (commonly used to visually evaluate the physical and mechanical properties of seedlings, and subsequently to grade the quality of seedlings), on acoustic speed has been published for seedlings. It is, hence, highly desirable to measure sound speed in seedlings with the application of an ultrasonic technique and to investigate the influencing factors on ultrasonic propagation speed in seedlings.

The poplar is a common plantation tree species in China, and poplar plantations provide an important raw material for papermaking, plywood, fiberboard, paper matches, sanitary chopsticks, and the packaging industry. The research work presented in this paper aimed to measure the ultrasonic propagation velocity in poplar (*Populus × euramericana cv.'74/76'*) seedlings, and to analyze the effect of various influencing factors such as wood density, MFA, growth days, and root-collar diameter on acoustic speed in seedlings. The results of this paper will provide some basic insights for the early screening of seedlings for their stiffness properties, using ultrasonic technology, and for developing a rapid evaluation method of poplar seedling wood quality in the future.

## 2. Materials and Methods

### 2.1. Materials

A total of 22 rows of poplar seedlings (*Populus × euramericana cv.'74/76'*) growing in the nursery base of Beijing Forestry University were planted at an initial spacing of 80 cm (row interval) × 30 cm (column interval), on April 10th. Then, 105 days later, 145 poplar seedlings numbered in sequence from P-001 to P-145 were randomly selected from the nursery base to conduct ultrasonic speed testing for the first time. After that, poplar seedlings were cut down, and a *l* (mm)-long specimen was cut from each green seedling using a portable electric saw. The extracted stem length, *l* is given by the equation

$$L = 15 \times d + 100 \tag{1}$$

where *d* (mm) is the root-collar diameter of the poplar seedling, and 100 mm is a reserved length of the specimen for density measurement. A total of 60 *l* (mm)-long specimens were obtained to perform acoustic resonance tests and density determination. These 60 specimens were immediately sealed by plastic wraps and directly transported to the wood nondestructive evaluation and testing laboratory in Beijing Forestry University, where they were kept in a condition room to maintain the green condition for poplar seedlings prior to acoustic resonance testing. A $15 \times d$ (mm)-long specimen and two 50-mm-long specimens (one from the top and the other one from the bottom) were cut from one l (mm)-long poplar seedling specimen. A total of 60 $15 \times d$ (mm)-long specimen were obtained to conduct acoustic resonance tests. It was found that $15 \times d$ (mm) is the optimal length for specimen to perform acoustic resonance tests, therefore, the ratio of length to diameter of specimen for acoustic resonance tests was 15 in this paper. Moreover, 60 50-mm-long specimens from the top of l (mm)-long poplar seedling specimen and 60 50-mm-long specimens from bottom, in total, were acquired and used to determine the green density of poplar seedlings by the water immersion method [22,23].

Additionally, two poplar seedlings, numbered P-146 and P-147, in the nursery were randomly chosen to determine the microfibril angle of seedlings. Five 30-mm-thick lines, named as A, B, C, D, and E, were marked on the poplar seedlings at the heights of 10, 50, 90, 130, and 170 cm above the base of seedling, respectively, as shown in Figure 1a. The ultrasonic speed between two discs, such as AB, BC, CD, or DE, was tested and recorded prior to being felled using a Fakopp Ultrasonic Timer (Sopron, Hungary) with a frequency of 90 kHz. After that, these two poplar seedlings were felled down and five discs, i.e., A, B, C, D, and E, were cut from each seedling. In order to obtain the specimens used for MFA measurement, at first, a 2-cm-wide strip was symmetrically taken from one disc along the pith from north to south direction. Then, a 2-cm-wide strip was symmetrically cut along the pith from east to west direction to obtain a specimen with a thickness of 0.2 cm (see Figure 1b). Finally, a total of 42 specimens measuring 2 (longitudinal) × 1.5 (tangential) × 0.15 (radial) cm obtained from the P-146 poplar seedling, and 28 specimens from the P-147 poplar seedling, were used to conduct the determination of the MFA of seedlings. These specimens were numbered as, e.g., P-146-A-1, where P-146 is the number of the poplar seedling, the letter A represents the number of the disc, and 1 means the number of the sampling position, as shown in Figure 1b. Specimens obtained from discs A, B, C, and D of P-146 poplar seedlings are shown in Figure 1c. Bruker D8 ADVANCE (Stuttgart, Germany), an X-ray diffractometer, was used in this paper to measure the MFA of specimens.

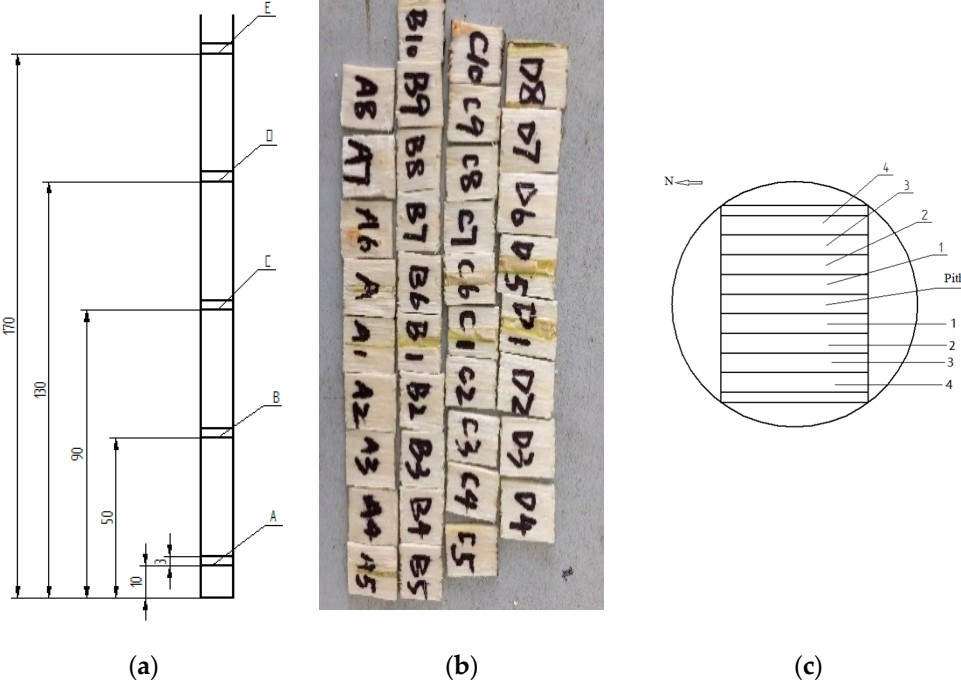

**Figure 1.** Schematic diagram of specimen preparation for MFA measurement: (**a**) disc sample position; (**b**) specimen sampling from cross-section; (**c**) specimens from discs A, B, C, and D of P-146 poplar seedlings.

*2.2. Methods*

2.2.1. Ultrasonic Method

A Fakopp ultrasonic timer, an ultrasonic instrument made in Hungary, was utilized to implement ultrasonic testing on poplar seedlings in situ. This ultrasonic tool is composed of two piezoelectric sensors with cables and one electronic box, as shown in Figure 2. These two sensors, a starter sensor and a receiver sensor, are placed on the same side of the seedling stem. The starter sensor, in field ultrasonic testing, was placed at a height of 200 mm above the ground. The receiver sensor was aligned with the starter sensor, and the distance between them was *L* (mm), namely measuring distance, seen in Figure 3a. Then, the ultrasonic propagation time for detection distance (*L*) from the starter sensor to the receiver sensor was displayed on the hand-held box. Three readings were recorded for each seedling, and then the average propagation time was used to calculate the corresponding ultrasonic velocity. Figure 3b shows the factual tests in situ using the Fakopp ultrasonic timer.

It should be noted that testing distance, *L*, was still uncertain due to the attenuation of ultrasound propagating in the seedling stems. Generally, a long distance between the starter and receiver sensors was necessary to fully reflect the wood properties of seedlings. However, if the test distance was too large, the ultrasonic pulse signal became weakened as the propagation distance increased. As a result, the receiver sensor did not trigger the timer, and thus no reading was recorded. Therefore, before ultrasonic testing, a pre-experiment was conducted to determine the proper detection distance between the two sensors when measuring ultrasonic velocity in poplar seedling stems. Six poplar seedlings with root-collar diameters ranging from 5 mm to 9 mm were selected to perform the pre-experiment using ultrasonic tools. In the pre-experiment, the detection distance, *L*, was set to a series of values, i.e., 65, 165, 265, 365, 465, 565, 665, 765, 865, and 965 mm. Then, the ultrasonic propagation time was

measured for these 10 various testing distances. Moreover, the ultrasonic velocity in the seedling stems was calculated according to Equation (2).

$$v_{\mathrm{u}} = \frac{L}{T - t_0} \times 1000 \tag{2}$$

where $v_{\mathrm{u}}$ is the ultrasonic velocity (m/s), $L$ is the detection distance (mm), $T$ is the transit time appearing on the ultrasonic instrument (s), and $t_0 = 6.1$ s is the time correction, which is the transit time inside the two sensors [24]. $T - t_0$ represents the true ultrasonic wave traveling time at the detection distance $L$ in the seedling stems. Ultimately, the proper detection distance for ultrasonic testing was determined based on wave velocities at different measuring distances.

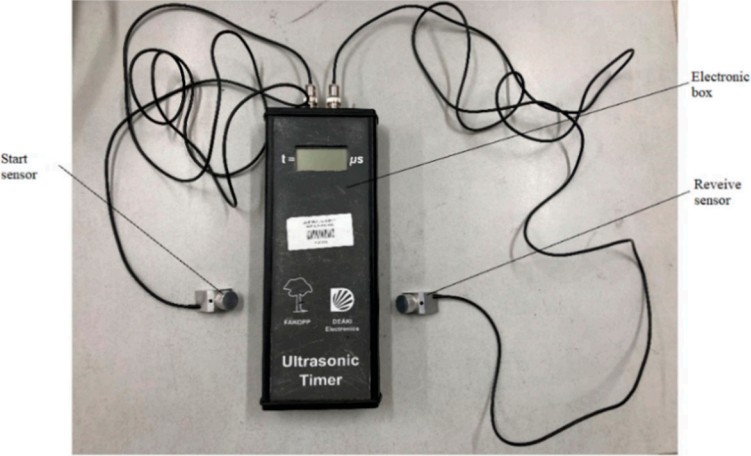

**Figure 2.** Fakopp ultrasonic timer.

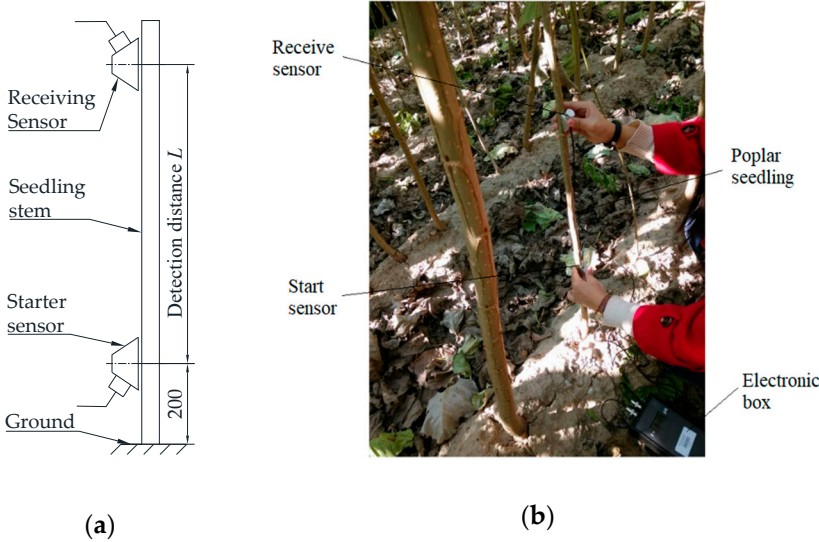

**Figure 3.** Ultrasonic propagation time measuring method: (**a**) Schematic diagram, (**b**) Field tests.

Once the proper detection distance (*L*) was identified, the ultrasonic propagation times of 145 seedling samples were measured, and then the ultrasonic propagation velocity of each seedling was calculated using Equation (2). To reduce the effect of environmental factors, the ultrasonic tests for the measurement of 145 seedlings were all completed on the same day. Simultaneously, the root-collar diameters of the seedlings were measured and recorded. A Vernier caliper was used in this step. Two diameters, perpendicular to each other such as east-west and north-south, at the starting point were measured and averaged. Then, the average was denoted as the final root-collar diameter. A total

of four timepoints of ultrasonic testing were performed on July 24, September 2, September 20, and November 5 in 2018, respectively—i.e., 105, 145, 165, and 209 days after the seedlings were planted.

### 2.2.2. Acoustic Resonance Method

Acoustic resonance techniques have been used to determine the stiffness of construction materials for a long time. In many respects, acoustic resonance techniques may obtain more useful results, as is evident by the correlations between time-of-flight measurement results on stems and resonance results on logs [25]. Acoustic resonance techniques are more accurate and repeatable than time-of-flight techniques [26], and generally do not require calibration and do not greatly depend on how the operator performs the measurements. The acoustic resonance test, a kind of stress-wave method, was performed in this part to be compared with the ultrasonic experiment described previously.

An acoustic resonance testing system (as shown in Figure 4), comprised of a hammer, two strings, a microphone (Type 2671, Brüel & Kjær, Copenhagen, Denmark), a signal amplifier (Type 1704, Brüel & Kjær, Copenhagen, Denmark) and a data acquisition card (Type USB-6218, National Instruments Corporation, Austin, TN, USA), was used to carry out acoustic resonance tests. As can be seen in Figure 3, the seedling specimens were hung with two strings, and the resonance signal generated by the hammer acted on one end of specimens, with a parallel direction. Then, the acoustic resonance signal was collected by the microphone, and afterwards transmitted to the amplifier and the data acquisition (DAQ) card. Finally, the signal was input to a computer for signal processing and analyzing. 60 15 × $d$ (mm)-long specimens cut from the seedling samples used for the ultrasonic tests were applied for the acoustic resonance tests to measure the periodic frequency of acoustic propagation in seedling specimens. Consequently, the acoustic velocity was determined through Equation (3).

$$V_a = 2l_af = 30df \tag{3}$$

where $v_a$ is acoustic velocity, $l_a$ is the length of specimen, $d$ is the root-collar diameter of poplar seedlings and $f$ is the frequency of resonance signal traveling between two ends of specimen. Furthermore, to compare with the modulus of elasticity (MOE) from the ultrasonic and acoustic resonance method, the dynamic MOE of these 60 seedling specimens derived from ultrasonic tests and acoustic resonance tests were calculated according to the determined density and measured transmitting velocity using the following equation.

$$E_d = \rho v^2 \tag{4}$$

where $E_d$ is the dynamic MOE of the seedling sample (Pa), $v$ is the wave velocity measured by either the ultrasonic test or the acoustic resonance test (m/s), and $\rho$ is the green density of the sample (kg/m$^3$). It should be noticed that the acoustic test was conducted on only one occasion, on 24 July in 2018, i.e., 105 days after the seedlings were planted.

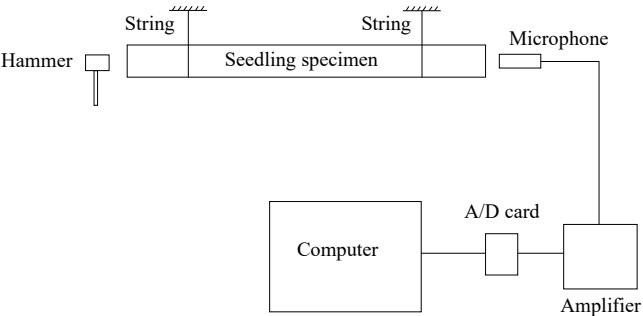

**Figure 4.** Test setup of the acoustic resonance method.

### 2.2.3. Density Measurement

A water immersion method was used to determine the green density of poplar seedlings. A total of 60 100-mm-long specimens were applied for the determination of their density. Firstly, two 50-mm-long specimens used for density measurement were cut from the top and bottom of each l-mm-long specimen, and then these two specimens were marked as top and bottom, respectively. After that, the mass of the top 50-mm-long specimen, marked as $m$ (g), was scaled and recorded using an electronic balance (see Figure 5a). A beaker with water inside was placed on the scale, and then a slim pin was vertically immersed in the water until the marked position, as shown in Figure 5b. Thus, the total mass of the beaker with the water and the immersed slim pin, marked as $m_0$ (g), was scaled and recorded by the electronic balance. Finally, the slim pin was inserted into the top 50-mm-long specimen at a depth of about 1 cm, and then this top specimen was immersed in the beaker until the same position marked on the slim pin, as seen in Figure 5c. Afterwards, the total mass of the beaker with the water, top specimen and immersed slim pin was scaled and recorded, and marked as $m_1$ (g).

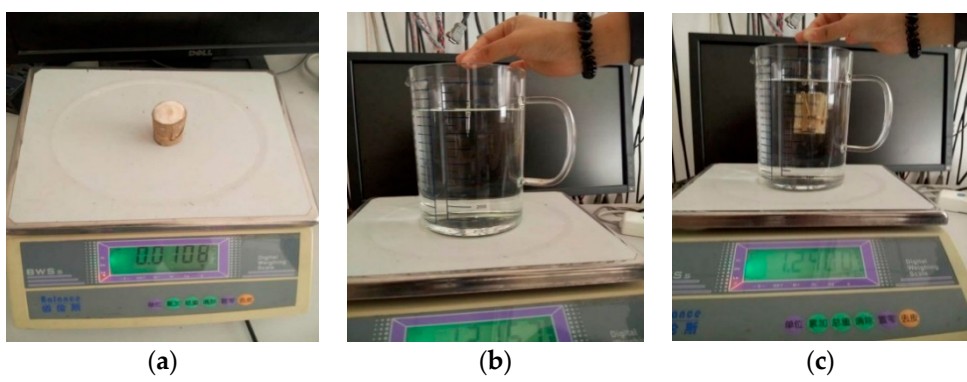

(**a**)            (**b**)            (**c**)

**Figure 5.** Density measurement for a 50-mm-long specimen: (**a**) the mass of $m$, (**b**) the mass of $m_0$, (**c**) the mass of $m_1$.

Since the density of water is 1 g/cm$^3$, the density of the top specimen can be calculated from the Equation (5). Similarly, the density of the bottom specimen can be determined by the above steps and calculated using Equation (5). Therefore, the average density value of the top and bottom specimens was the green density of a poplar seedling specimen. It should be noted that the density measurement method described in this paper only works for wood which has a lower density than water. Another denser liquid, such as mercury, can be used for certain sapwood specimens that have greater densities than water.

$$\rho = \frac{m}{(m_1 - m_0)\cdot 1 \text{ g/cm}^3} \tag{5}$$

## 3. Results and Discussion

### 3.1. Proper Detection Distance for Ultrasonic Tests

Figure 6 shows the relationship between detection distance and ultrasonic velocity for six different seedling samples, i.e., P002, P004, P005, P006, P007, and P008. The root-collar diameters of these six seedling samples, in sequence, were 8.1, 5.9, 6.8, 9.0, 7.1, and 7.2 mm. It can be obviously observed from Figure 6 that no ultrasonic velocity values were obtained when the detection distance was over 765 mm for the P004 seedling. A similar result was also found in the P005 seedling when the detection distance was greater than 865 mm. This may be due to smaller root-collar diameters in the P004 and P005 seedlings (5.9 mm and 6.8 mm, respectively) compared with the other four seedling samples. Ultrasonic propagation time could not be measured as the root-collar diameter was too small.

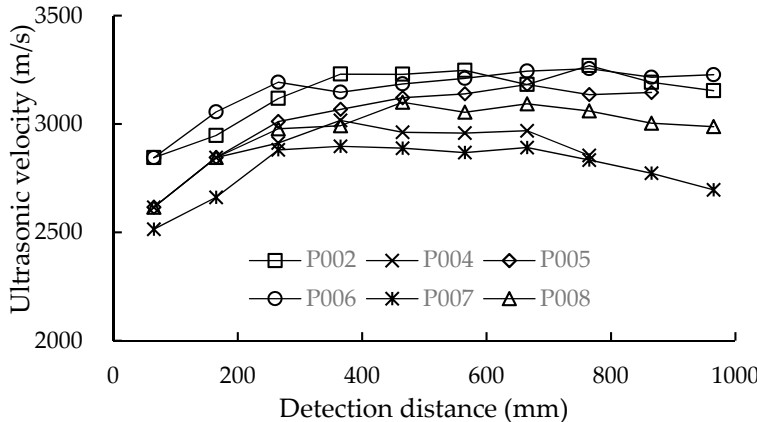

**Figure 6.** Relationship between detection distance and ultrasonic velocity.

It can be seen from Figure 6 that the ultrasonic velocities of the six seedlings all significantly increased when the detection distance changed from 65 mm to 265 mm. For the P004 and P007 seedlings, the ultrasonic velocities both remained basically stable when the detection distance increased from 265 mm to 665 mm, and then linearly decreased once the test distance went beyond 665 mm. For the P002 and P008 seedlings, the ultrasonic velocity generally remained steady when the detection distance was larger than 265 mm. However, for P005 and P006, the ultrasonic velocity slightly increased when the detection distance was larger than 265 mm. Therefore, the proper detection distance for the ultrasonic test should be chosen from 265 mm to 665 mm. Considering the convenience and feasibility of experimental tests, the detection distance used in this paper was ultimately set to 365 mm. The ultrasonic wave speeds shown in the following were all measured at a test distance of 365 mm.

*3.2. Ultrasonic Velocity and Root-Collar Diameters of Poplar Seedlings*

The measured ultrasonic velocities and root-collar diameters of poplar seedlings from tests conducted on July 24, September 2, September 20, and November 5 in the same year, 2018, respectively, are given in Table 1. The values in brackets besides average root-collar diameter and velocity are standard deviations. The average root-collar diameter of 145 seedling samples obtained from four different test dates was 15.98 mm, 20.88 mm, 22.06 mm, and 22.60 mm, respectively. A fast growth rate was observed in average root-collar diameter from July 24 to September 2. In contrast, a lower growth rate was found from September 20 to November 5. This means that poplar seedlings may have a fast growth rate from July to September, and a low growth rate from September to November. Moreover, a low growth rate was also found from September 2 to September 20 due to the shorter growth days.

Ultrasonic velocity, apparently, was increased with growth days. Moreover, in the first (July 24), second (September 2), third (September 20), and fourth (November 5) ultrasonic tests, the maximum values of average ultrasonic velocity—i.e., 1995, 2518, 2703, and 2953 m/s, respectively—were all found in poplar seedlings with a root-collar diameter between 10 mm and 20 mm. The minimum values of average velocity from the first ultrasonic test were found in poplar seedlings with a root-collar diameter between 20 mm and 30 mm. However, for the second and third tests, the minimum values of average velocity were presented in poplar seedlings with a root-collar diameter over 30 mm. Meanwhile, the poplar seedlings with a root-collar diameter of less than 10 mm had the minimum values of average ultrasonic velocity.

**Table 1.** Results of ultrasonic velocity and root-collar diameter for 145 poplar seedlings.

| Date | Root-Collar Diameter (mm) | Number | Average Root-Collar Diameter (mm) | Maximum Root-Collar Diameter (mm) | Minimum Root-Collar Diameter (mm) | Average Velocity (m/s) | Maximum Velocity (m/s) | Minimum Velocity (m/s) |
|---|---|---|---|---|---|---|---|---|
| 24 July | d ≤ 10 | 15 | 8.4(1.5) | 10.0 | 4.9 | 1921(219) | 2241 | 1675 |
| | 10 < d ≤ 20 | 101 | 15.4(2.9) | 20.0 | 10.3 | 1995(123) | 2269 | 1680 |
| | 20 < d ≤ 30 | 29 | 21.9(1.4) | 25.0 | 20.1 | 1919(122) | 2087 | 1748 |
| 2 September | d ≤ 10 | 5 | 8.0(1.5) | 9.4 | 5.7 | 2435(116) | 2520 | 2283 |
| | 10 < d ≤ 20 | 56 | 15.4(2.6) | 20.0 | 10.2 | 2518(153) | 2747 | 2136 |
| | 20 < d ≤ 30 | 79 | 24.9(2.7) | 29.9 | 20.2 | 2315(122) | 2707 | 1996 |
| | d > 30 | 5 | 31.6(1.1) | 33.3 | 30.5 | 2145(88) | 2283 | 2041 |
| 20 September | d ≤ 10 | 5 | 8.0(1.6) | 9.7 | 5.7 | 2509(137) | 2667 | 2404 |
| | 10 < d ≤ 20 | 49 | 15.4(2.6) | 19.5 | 10.2 | 2703(125) | 2923 | 2327 |
| | 20 < d ≤ 30 | 73 | 25.1(2.7) | 29.6 | 20.1 | 2484(134) | 2833 | 2228 |
| | d > 30 | 18 | 31.8(1.6) | 35.0 | 30.1 | 2328(57) | 2404 | 2188 |
| 5 November | d ≤ 0 | 5 | 8.4(1.4) | 9.9 | 6.8 | 2669(281) | 2855 | 2555 |
| | 10 < d ≤ 20 | 48 | 15.6(2.6) | 19.9 | 10.2 | 2953(107) | 3234 | 2707 |
| | 20 < d ≤ 30 | 71 | 25.4(2.8) | 29.8 | 20.5 | 2884(133) | 3353 | 2629 |
| | d > 30 | 21 | 32.5(1.7) | 36.3 | 30.1 | 2741(92) | 2971 | 2573 |

Figure 7 shows the results of the statistical analysis for ultrasonic velocities measured on July 24 and November 5. It can be clearly found from Figure 7 that the ultrasonic velocities of poplar seedlings measured on November 5 were significantly greater than those tested on July 24, maybe due to the longer growth days or the changes in wood density throughout the growing season. In addition, as shown in Figure 7a, the majority of ultrasonic velocities in the first test (July 24) were concentrated near the average velocity (1974 m/s) and ranged from 1825 m/s to 2025 m/s. Moreover, for the fourth test (November 5), the majority of ultrasonic velocities were likewise concentrated near the average ultrasonic velocity (2881 m/s), but ranged from 2680 m/s to 3160 m/s.

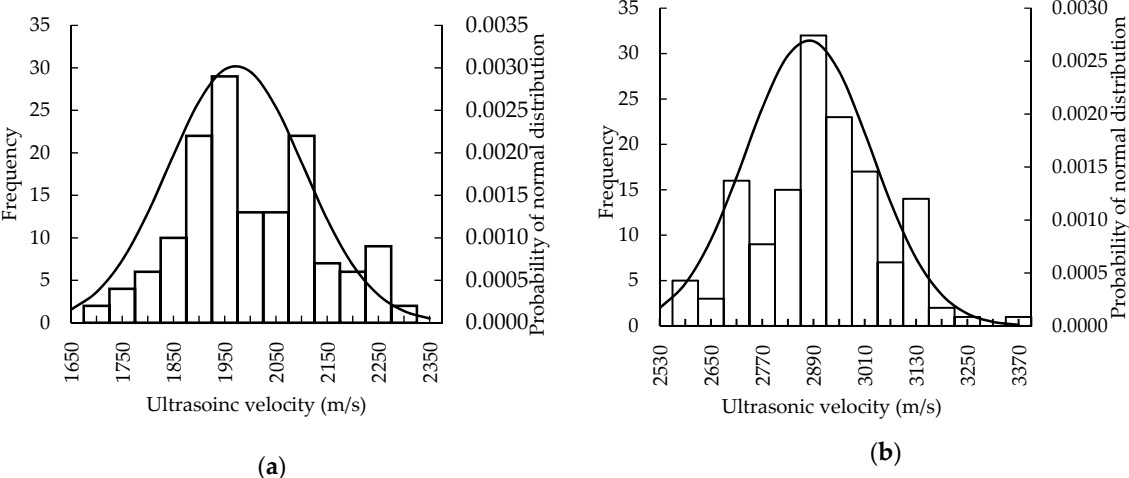

(**a**)   (**b**)

**Figure 7.** Results of statistical analysis for ultrasonic velocities: (**a**) test on July 24; (**b**) test on November 5.

### 3.3. Relationships Between Ultrasonic Velocity and Influencing Factors

### 3.3.1. Growth Days

The effect of growth days on ultrasonic speed in poplar seedlings was analyzed, and the relationships between ultrasonic velocity and growth days are illustrated in Table 2 and Figure 8. The results of the statistical analysis for ultrasonic velocity and growth days are summarized in Table 2. The upper limit of velocity is the wave speed value at the position of positive 3σ in the ultrasonic velocity probability distribution histogram. Conversely, the lower limit of velocity is the value at the position of negative 3σ in the ultrasonic velocity probability distribution histogram. As can be seen in Table 2, the average ultrasonic velocities of poplar seedlings were 1972, 2365, 2540, and 2879 m/s, corresponding to 105, 145, 165, and 209 growth days. The ultrasonic speed of poplar seedlings increased with growth days, within 209 growing days. This result means that the growth days may play a positive role in the ultrasonic velocity of poplar seedlings. Similarly, the growth days have a positive influence on the upper limit of velocity and the lower limit of velocity. The upper limit wave speed increased from 2368 to 3323 m/s, and the lower limit wave speed increased from 1576 to 2434 m/s, when growth days increased from 105 to 209 days.

**Table 2.** Results of statistical analysis for ultrasonic velocity and growth days.

| Date | Number of Trees | Growth Days | Average Velocity | Standard Deviation | Upper Limit of Velocity | Lower Limit of Velocity |
| --- | --- | --- | --- | --- | --- | --- |
| | | | (m/s) | | (m/s) | (m/s) |
| 24 July | 145 | 105 | 1972 | 132 | 2368 | 1576 |
| 2 September | 145 | 145 | 2365 | 166 | 2865 | 1866 |
| 20 September | 145 | 165 | 2540 | 183 | 3087 | 1992 |
| 5 November | 145 | 209 | 2879 | 148 | 3323 | 2434 |

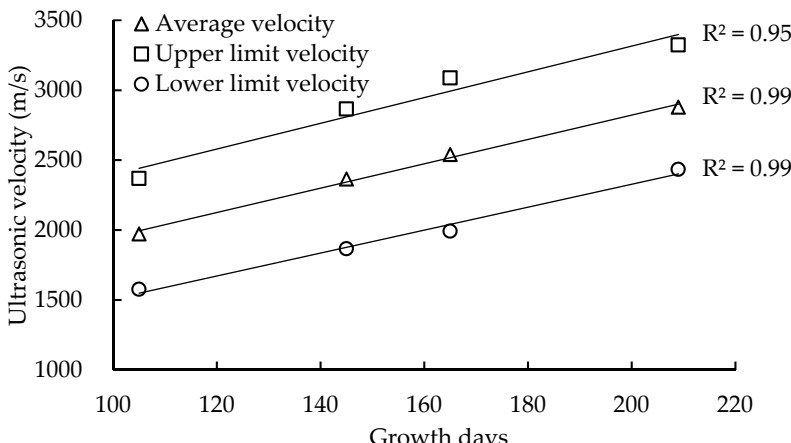

**Figure 8.** Relationship between ultrasonic velocity and growth days.

Figure 8 shows the results of the correlation analysis between the ultrasonic velocities and seedling growth days. It can be seen from Figure 8 that the average velocity, upper limit of velocity and lower limit of velocity were all linearly increased with growth days, within 209 growing days. There are good correlations between these three kinds of ultrasonic velocities and growth days, and the correlations ($R^2$) between the average velocity, upper limit of velocity, lower limit of velocity, and the growth days were 0.99, 0.95, and 0.99, respectively. There was a dramatic difference in the ultrasonic velocities of seedlings at different growth days. This is may be due to the fact that the mechanical properties of poplar seedlings gradually become better as growth days increase, resulting in an increase in ultrasonic wave velocity. Therefore, it could be predicted that the ultrasonic velocity of seedlings would continually increase with growth days due to the underlying changes in density or other wood properties. Growth days, thus, may play an important role in the ultrasonic speed of early stage poplar seedlings, especially within 209 growth days. However, it should be noted that the ultrasonic velocity is likely changing due to the underlying changes in density or other wood properties, and not directly due to more growing days. Underlying factors such as moisture content, MFA, wood density, and other wood properties are also likely to have a role in the development of mechanical properties that affect the ultrasonic velocity.

### 3.3.2. Root-Collar Diameter

It is necessary to investigate the effect of root-collar diameter on ultrasonic velocity and then to determine the relationship between root-collar diameter and ultrasonic speed. In general, the root-collar diameter of seedlings increased with growth days (i.e., the age of seedlings). As showed in Figure 8, the ultrasonic velocity in seedlings linearly increased with increasing growth days. Thereby, it could be speculated that ultrasonic speed may increase with increasing root-collar dimeter of seedlings. However, from the relationships between the ultrasonic velocity and root-collar diameter of poplar seedlings provided in Figure 9, it can be observed that for the four different test dates, the ultrasonic velocity kept relatively stable as the root-collar diameter of poplar seedlings increased. There was no significant correlation between ultrasonic velocity and root-collar diameter in this paper. In other words, it seems that ultrasonic speed does not increase with root-collar diameter.

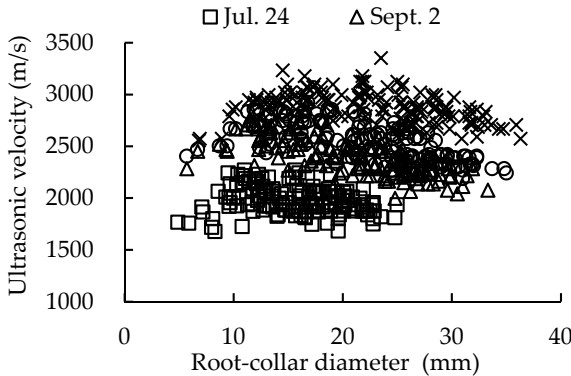

**Figure 9.** Relationship between the ultrasonic velocity and root-collar diameter of seedlings.

In addition, it was found that the overall ultrasonic velocities measured on November 5 were the highest among the four timepoints of ultrasonic tests, while the test on July 24 obtained the lowest velocity. Ultrasonic velocity was overall increased with growth days, which is consistent with the results shown in Figure 8. Therefore, the root-collar diameter of seedlings did not show a significant effect on ultrasonic velocity in this work. More data from the same or diverse seedlings species, however, still need to be acquired to further verify the effect of root-collar diameter on ultrasonic speed and the relationship between them.

### 3.3.3. Density

Wood density is also often utilized to assess the mechanical and physical performance of seedlings, and then to classify the quality of seedlings. Accordingly, it is essential to learn the impact of density on ultrasonic velocity in poplar seedlings and the correlation between wood density and ultrasonic speed.

Figure 10 shows the relationship between the density of seedlings and ultrasonic velocity. It can be seen from Figure 10 that ultrasonic velocity overall tends to decrease with increasing density of seedlings. However, a low value of correlation ($R^2 = 0.26$) was found between ultrasonic velocity and density. Therefore, the relation between ultrasonic speed and the density of seedlings was not significant. This poor relation may be contributed to by the fact that ultrasound speed not only depends on density, but also on other factors such as MFA, grain angle, and the age of seedlings.

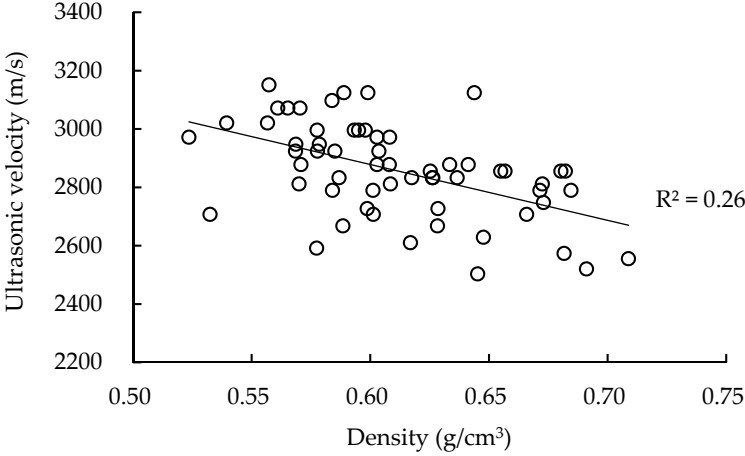

**Figure 10.** Relationship between the ultrasonic velocity and density of poplar seedlings.

The effect of wood density on acoustic speed has been investigated and reported in many studies. However, two contrasting results were found in the relationship between density and acoustic speed. Some studies reported a negative relationship between wood density and acoustic velocity. Isik et al., for instance, found a low but negative correlation for air-dry density and acoustic velocity in *Pinus taeda*,

and the corresponding correlation coefficient was −0.2 [27]. Hasegawa et al. showed that the ultrasonic wave velocities of Japanese cedar and Japanese cypress both linearly decreased with air-dry density, and the correlation coefficients were −0.83 and −0.74, respectively [28]. However, other studies indicated a positive relation between wood density and acoustic speed. Krauss et al., for example, reported a low but positive relation for density and ultrasonic velocity in Scots pine, and the correlation was 0.15 [29]. Chen et al. and Lachenbruch et al. both found a moderate and positive relationship for green density and ultrasonic speed, and the correlations were 0.46 in Norway spruce and 0.33 in Douglas fir, respectively [12,30]. Moreover, Blackburn et al. and Ribeiro et al. found that acoustic speed greatly increased with wood basic density, and the correlations were 0.75 for *Eucalyptus nitens* and 0.84 for *Pinus taeda* [31,32]. The results of the present paper were basically in accordance with those reported in Isik et al.'s work. Therefore, density may have an influence on ultrasonic speed. More efforts definitely need to be put into the investigation of the impact of density on ultrasonic speed and the correlations between ultrasonic velocity and density in identical or different seedling species. It may help to early select the better-quality seedlings with high average values of stiffness, if the effect of wood density on ultrasonic velocity were comprehensively understood.

### 3.3.4. Microfibril Angle

The microfibril angle of the S2 cell wall layer is an important parameter of wood. Many studies have showed that the microfibril angle was highly related to the mechanical properties of wood and acoustic speed, for logs and lumbers [33]. The stiffness—i.e., modulus of elasticity—and the acoustic velocity of wood decreased as the microfibril angle increased. Therefore, it is necessary to figure out the effect of microfibril angle on ultrasonic velocity in poplar seedlings and the relationship between MFA and ultrasonic speed.

Microfibril angles measured at different positions (1, 2, 3, 4, ... as shown in Figure 1b) of one disc were averaged and used as the microfibril angle of this disc. Then, the average of the microfibril angle of discs A and B is taken as the microfibril angle between disc A and B. Similarly, the microfibril angles between B and C, C and D, and D and E can be obtained. Thus, the ultrasonic speeds for sections AB, BC, CD, and DE and their corresponding microfibril angles were used to analyze the correlations between them. Figure 11 shows the relationship between the microfibril angle of seedlings and ultrasonic velocity. It can be seen from Figure 11 that ultrasonic velocity significantly decreased with increasing microfibril angle of seedlings. The correlation ($R^2$) between ultrasonic velocity and MFA for poplar seedling P-146 was 0.69, which is lower than that for poplar seedling P-147 ($R^2 = 0.89$). Therefore, the relation between ultrasonic speed and MFA is of great interest for these two seedlings, and MFA may have an impact on ultrasonic speed in poplar seedlings. More seedling specimens are definitely needed to verify this relationship and confirm the effect of MFA on ultrasonic velocity.

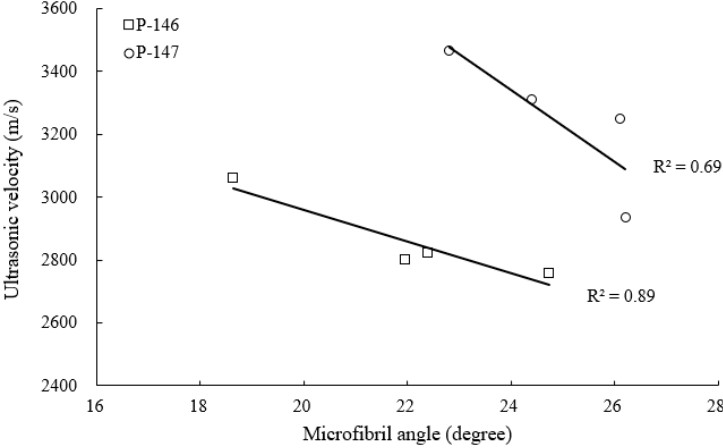

**Figure 11.** Relationship between the ultrasonic velocity and microfibril angle of poplar seedlings.

MFA has been reported to have a significant and negative relation with acoustic velocity for wood materials such as logs and lumbers in many studies. Krauss et al., for example, found a negative relationship ($R^2$ = 0.71) between the ultrasonic wave velocity and MFA of Scots pine [29]. Chen et al. reported a very high but negative relationship ($R^2$ = 0.98) between acoustic velocity and MFA in Norway spruce [30]. Isik et al. and Lachenbruch et al. both observed that acoustic velocity greatly decreased with increasing MFA, and the correlations were 0.70 and 0.69 for *Pinus taeda* and Douglas fir, respectively [12,27]. Moreover, Hasegawa et al. showed that for Japanese cedar, the correlation between ultrasonic wave velocity and MFA was 0.90, and 0.82 for Japanese cypress. They suggested that MFA greatly affects the ultrasonic wave velocity in softwood [28]. The results of the present paper are consistent with their reported results.

### 3.4. Comparison with the Results of Acoustic and Ultrasonic Tests

Figure 12 shows the relationship between ultrasonic velocity and acoustic resonance velocity in the 60 poplar seedlings. The average ultrasonic velocity and acoustic velocity for these 60 poplar seedlings were 3114.8 m/s and 2856.7 m/s, respectively. The standard deviations of ultrasonic velocity and acoustic velocity were 159.53 and 164.68, respectively. The average ultrasonic velocity was approximately 9.1% (i.e., 260 m/s) higher than the average acoustic velocity. In addition, it can be seen from Figure 12 that there was a significant relationship between ultrasonic velocity and acoustic velocity, and the correlation ($R^2$) was 0.81. Similar results have been reported by other researchers as well [1,23]. The acoustic resonance method is generally recognized as a reliable and accurate method for measuring the sound speed of wood material, such as logs and lumbers. Therefore, the prominent relationship between ultrasonic and acoustic velocity may indicate that the ultrasonic method can be used to measure the ultrasonic sound speed of poplar seedlings.

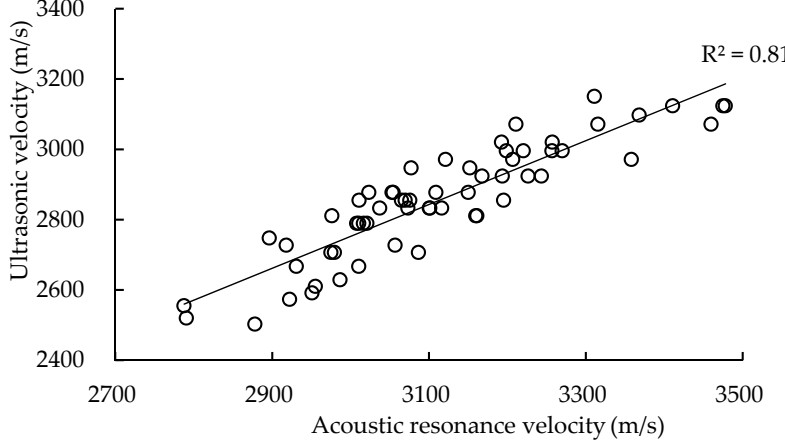

**Figure 12.** Relationship between ultrasonic velocity and acoustic resonance velocity.

Figure 13 presents the relationship between the dynamic MOE results obtained from ultrasonic and acoustic tests carried out in 60 seedlings. The average dynamic MOE values derived from the ultrasonic and acoustic tests for these 60 poplar seedlings were 5.92 and 4.98 GPa, respectively. The standard deviations of dynamic MOE derived from the ultrasonic and acoustic methods were 0.48 and 0.36, respectively. The average dynamic MOE from the ultrasonic test was approximately 18.87% (i.e., 0.94 GPa) higher than that from the acoustic test. Additionally, it can be observed from Figure 13 that there was a good relationship between dynamic MOE from the ultrasonic test and the acoustic test, and the correlation ($R^2$) was 0.75. It is well known that the dynamic elastic modulus measured by the acoustic method can be well used to predict the static elastic modulus of wood. Therefore, this means that the dynamic elastic modulus measured by the ultrasonic method may be able to be used for predicting the static elastic modulus of wood, especially for young seedlings, due to the noticeable relationship between dynamic MOE results from ultrasonic and acoustic tests. However,

there is still a lot of work that needs to be done to investigate whether the ultrasonic method could be potentially utilized to evaluate the quality of young seedlings. Moreover, if ultrasound could be applied to the early selection of seedlings, the quality of standing trees and wood-based products may be improved because of the good quality of the young seedlings. Although good quality young seedlings do not guarantee good quality standing trees, they are a good start to cultivate the high properties of plantation trees.

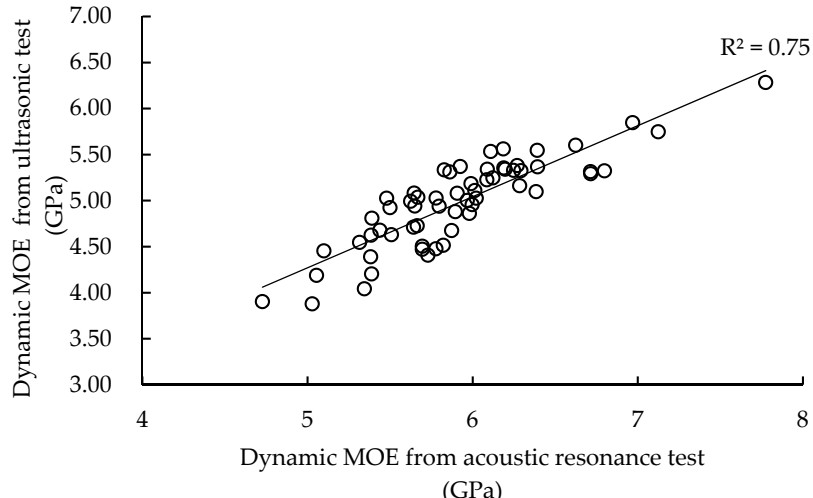

**Figure 13.** Relationship between dynamic MOE values from ultrasonic and acoustic resonance tests.

## 4. Conclusions

The aim of this study was to investigate the application of ultrasonic speed measurement to poplar (*Populus × euramericana cv.'74/76'*) seedlings, a common plantation species in China, and to gain some basic insights for the future early selection of poplar seedlings with high wood quality. The conclusions are as follows:

- The proper detection distance for the ultrasonic method to be applied to poplar seedlings is ranged from 265 mm to 665 mm, and 365 mm was used in this study.
- There were good correlations ($R^2 = 0.99$) between the average ultrasonic velocities and growth days. Ultrasonic speed increased with growth days, within 209 growing days. However, almost no relationship was found between the ultrasonic velocities and the root-collar diameters of seedlings, i.e., ultrasonic speed does not seem to increase with increasing root-collar diameter.
- Even though ultrasonic velocity, in general, decreases with increasing density, the density of seedlings showed a weak influence on ultrasonic speed due to the low correlation between them ($R^2 = 0.26$). However, ultrasonic velocity significantly decreased with the increasing microfibril angle of seedlings. The relations between ultrasonic speed and MFA are of great interest for the two sample seedlings, and MFA may have an impact on ultrasonic speed in poplar seedlings. More seedling specimens are definitely needed to verify this relationship and confirm the effect of MFA on ultrasonic velocity.
- There was a significant relationship between ultrasonic velocity and acoustic velocity, and a similar result was also found in the dynamic MOE values derived from the acoustic resonance test and the ultrasonic test, respectively.
- Other influencing factors that were excluded in this paper—such as MFA, temperature, and moisture content—need to be studied in future research to investigate their effect on ultrasonic velocity in seedlings.

**Author Contributions:** F.L. rewrote and revised the manuscript as well as conducted part of data analysis work. P.X. conducted part of analysis work and wrote the original draft of the manuscript. H.Z. and C.G. supervised the

research team and provided some ideas to research as well as revised the manuscript. D.F. performed most of test and analysis work. X.W. revised the manuscript.

**Funding:** This project was supported by "the Fundamental Research Funds for the Central Universities (no. BLX201817)", China Postdoctoral Science Foundation (no. 2018M641225) and the National Natural Science Foundation of China (no. 31328005).

**Acknowledgments:** The authors wish to thank Jianhua Hao for his grateful assistance of planting the seedlings.

**Conflicts of Interest:** The authors declare no conflict of interest. The funders had no role in the design of the study; in the collection, analyses, or interpretation of data; in the writing of the manuscript, or in the decision to publish the results.

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
