# Peer review of "Use of Time-of-Flight Ultrasound to Measure Wave Speed in Poplar Seedlings"

_forests, doi:10.3390/f10080682_

Round 1

Reviewer 1 Report

I think that this is a useful paper which moves along research into using acoustic and other methods to select for stiffness in seedlings.

I can see that there was a lot of hard work behind this paper, and I congratulate the authors for putting the paper together.

The challenge I found with the paper is that the English needs significant editing to remove the grammar mistakes, to clarify some sentences, to use correct English words, and to eliminate the occasional overly formal terms (which make the paper read like a legal document at times).

General areas for consideration are:

Abstract (and other areas): sometimes the term ‘correlation’ is used for R2 and sometimes ’coefficient of determination’. Please just use one of these terms.

Citations within the body of the paper need checking: I found my work was being referred to as ‘Grant et al.’ when it should be ‘Emms et al.’

Section 2.2.1: Why were the samples extracted on the given days (105,145,165,209) ? They aren’t evenly spaced periods.

Section 2.2.3: It is not clear in this section whether the specimens are dry or green wood samples. One presumes green.

Why cut the 100mm section into 2? Surely measuring the density on just one 100mm section will produce more accurate results, compared to averaging results from 2 sections.

I would note to the readers that the density method explained in the paper only works for densities less than water density. Many sapwood specimens of different tree species will have densities greater than water. (or maybe you could use another, denser liquid, like mercury…. or, perhaps not).

Section 3.2: I feel that a lot of the text is redundant in this section, and it is enough to look at the table and figure. I think that everything from line 256 (starting at ‘As can be seen,…’) to line 280 can be removed, and everything from line 284 (starting at ‘In addition,’) to line 287 can be removed. In any case, make sure the text isn't just repeating what is in the table and figure.

Section 3.3.1: I am not sure what ‘growth days’ (and ‘growing days’) really means, and whether it is a standard term (maybe it is – but I haven’t noted it before). I would have thought ‘seedling age’ is okay. In fact, line 324 sort of clarifies this in the next section. Either way, ‘growth days’ (and ‘growing days’) confused me at the beginning. I must say, I initially though that ‘growth days’ meant the days the seedlings was in a growth phases, as opposed to being dormant.

In figure 8 I’m not sure how valid, or rather accurate, R2 is for four points, especially since two measurement are quite close to each other. Similarly for figure 11.

Section 3.3.3: Lines 355 to 366: One must be clear whether these studies are talking about basic density, air dry (to a certain MC) density, or green density. In this paper, green density is being used (I think). If all other factors (MFA etc.) are constant increasing density decreases acoustic velocity. This is more likely for green wood since the free water density may change in seedlings depending on the amount of sapwood.

More specific points of concern:

Line 39: I would expect that plantation trees have lower stiffness etc. I think the author means to say that timber/lumber needs to have the listed properties and plantation trees need to produce more such timber.

Line 109: Need to define 'l better. So add “The extracted stem length, l, is given by the equation“.

Line 349: The sentences ‘Therefore, the relation between ultrasonic speed and density have not a great interest. The poor relation may be contributed to the fact that ultrasound speed, as well known does not depends only on density, but also on the other factors such as MFA, grain angle and the age of seedlings.’ don’t make a lot of sense, and I’m not sure what the authors mean.

Conclusions: Line 438: I thought that separations from 265 to 665mm were acceptable, not just 365mm.

Author Response

Response to Reviewer 1 Comments

Point 1: I think that this is a useful paper which moves along research into using acoustic and other methods to select for stiffness in seedlings. I can see that there was a lot of hard work behind this paper, and I congratulate the authors for putting the paper together. The challenge I found with the paper is that the English needs significant editing to remove the grammar mistakes, to clarify some sentences, to use correct English words, and to eliminate the occasional overly formal terms (which make the paper read like a legal document at times)..

Response 1: Special thanks to you for the affirmation of our scientific research work and paper. We have carefully revised the manuscript based on your comments and recommendations. This manuscript has carefully undergone English language editing by MDPI, which are highlighted in blue colour in the manuscript. The text has been checked for correct use of grammar and common technical terms.

Point 2: Abstract (and other areas): sometimes the term ‘correlation’ is used for R2 and sometimes ’coefficient of determination’. Please just use one of these terms.

Response 2: This is a good commendation. The term used for R2 has been changed to “correlation” in the whole manuscript.

Point 3: Citations within the body of the paper need checking: I found my work was being referred to as ‘Grant et al.’ when it should be ‘Emms et al.’.

Response 3: Thank you so much for pointing out this problem. We have carefully checked citations within the body of the paper and errors was corrected in the whole manuscript, especially changing “Grant et al.” to “Emms et al.”

Point 4: Section 2.2.1: Why were the samples extracted on the given days (105,145,165,209) ? They aren’t evenly spaced periods.

Response 4: This is a good question. Actually, a perfect and even spaced period was not perfomed for ultrasonic measurement in this paper because of the weather condition and human factors. In addition, growth days rather than even spaced period was the influencing factor on ultrasonic velocity discussed in this article, therefore, it is not necessary to extract the samples on even spaced periods though it would be better to adopt an even spaced period.

Point 5: Section 2.2.3: It is not clear in this section whether the specimens are dry or green wood samples. One presumes green.

Response 5: The measured densities of specimens in section 2.2.3 were green density. Detailed descriptions were added in the manuscript.

Point 6: Why cut the 100mm section into 2? Surely measuring the density on just one 100mm section will produce more accurate results, compared to averaging results from 2 sections.

Response 6: Thanks for your recommendations. There are some problems about describing the sampling methods for specimens used for density determination. Actually, two 50mm-long specimens used for density determination was cut from the top and bottom of l (cm)-long seedling specimen, respectively, in order to represent the average density of one poplar seedlings. These two 50mm-long specimens for density measurement was not cut from a whole 100mm section. This confusion have been solved and revised in the manuscript.

Point 7: I would note to the readers that the density method explained in the paper only works for densities less than water density. Many sapwood specimens of different tree species will have densities greater than water. (or maybe you could use another, denser liquid, like mercury…. or, perhaps not).

Response 7: This is a very good recommendation. As reviewer mentioned, the density measured method used in this article only works for wood which has a lower density than water. This has been in detail described in the text.

Point 8: Section 3.2: I feel that a lot of the text is redundant in this section, and it is enough to look at the table and figure. I think that everything from line 256 (starting at ‘As can be seen,…’) to line 280 can be removed, and everything from line 284 (starting at ‘In addition,’) to line 287 can be removed. In any case, make sure the text isn't just repeating what is in the table and figure.

Response 8: Thanks for your recommendation. The section 3.2 has been revised and some repeat words have been removed.

Point 9: Section 3.3.1: I am not sure what ‘growth days’ (and ‘growing days’) really means, and whether it is a standard term (maybe it is – but I haven’t noted it before). I would have thought ‘seedling age’ is okay. In fact, line 324 sort of clarifies this in the next section. Either way, ‘growth days’ (and ‘growing days’) confused me at the beginning. I must say, I initially though that ‘growth days’ meant the days the seedlings was in a growth phases, as opposed to being dormant.

Response 9: As reviewer mentioned, the term “growth days” and “growing days” used in this article in fact was referred to seedling age rather than the days that the seedlings was in a growth phase. This was clarified in the Section 3.3.2.

Point 10: In figure 8 I’m not sure how valid, or rather accurate, R2 is for four points, especially since two measurement are quite close to each other. Similarly for figure 11.

Response 10: Thanks for your comments. Generally, four data points were basically satisfied the criteria to conduct the linear fitting using least square method. Only two measurements for P-146 in Figure 11 are found approach velocity values, therefore, the coefficient of determination R2 shown in Figure 8 and 11 should be valid and accurate according to the results of linear correlation analysis.

Point 11: Section 3.3.3: Lines 355 to 366: One must be clear whether these studies are talking about basic density, air dry (to a certain MC) density, or green density. In this paper, green density is being used (I think). If all other factors (MFA etc.) are constant increasing density decreases acoustic velocity. This is more likely for green wood since the free water density may change in seedlings depending on the amount of sapwood.

Response 11: This is a very good recommendation. The type of density of these studies referred in the Section 3.3.3 has been specifically added in the manuscript. And the green density was exactly used in this paper.

Point 12: Line 39: I would expect that plantation trees have lower stiffness etc. I think the author means to say that timber/lumber needs to have the listed properties and plantation trees need to produce more such timber.

Response 12: Yes, correct. If plantation trees have lower stiffness, more trees need to be cut down to produce the listed properties, therefore, the outturn of plantation trees was lower.

Point 13: Line 109: Need to define 'l better. So add “The extracted stem length, l, is given by the equation”.

Response 13: The definition of “l” has been added into the text.

Point 14: Line 349: The sentences ‘Therefore, the relation between ultrasonic speed and density have not a great interest. The poor relation may be contributed to the fact that ultrasound speed, as well known does not depends only on density, but also on the other factors such as MFA, grain angle and the age of seedlings.’ don’t make a lot of sense, and I’m not sure what the authors mean.

Response 14: These sentences have been revised in the manuscript. We would like to express that the relation between ultrasonic speed and the density of seedlings was not significant and this may be contributed to the fact that ultrasound speed does not only depends on density, but also on the other factors such as MFA, grain angle and the age of seedlings.

Point 15: Conclusions: Line 438: I thought that separations from 265 to 665mm were acceptable, not just 365mm.

Response 15: Thanks for your comments. Conclusions in line 438 have been revised. As reviewer mentioned, the proper detection distance of ultrasonic method for poplar seedlings was ranged from 265mm to 665mm, though 365 mm was used to measure the ultrasonic speed of seedlings in this study.

Reviewer 2 Report

Will need moderate but essential changes to English language issue.  Examples are:

line #15 not necessary to capitalize tree name Poplar - lower case as used in line 101 better.  If you agree needs to be changed throughout article.

line 34 - confusing last few words "and in future" ?

line 39 -  "forests are expected to"  , drop "greatly"

line 41 - ", thus yielding lumber, "

line 46 - "many studies showing that it is" use the gerund form

line 51 - "it was worth developing"  delete infinitive "of" 

line 52 - use italics for P. radiata as ou hae done elsewhere in the article

line 75 - "more efforts are still needed to find a proper" delete "to put into"

line 85, "As previously mentioned" also use italics for P radiata

line 108 "fell down"  ???  do you mean cut down? note lines 123, 124

line 114 - simply say kept and delete "them"

line 130 - "like as" is incorrect why not just say either as or like?

line 142, 150 - unless the journal has different rules, one usually italicizes words like "in situ"

line 211 - "A water immersion"  - articles such as  "a" or "the" are missing from certain nouns throughout. Spell check should help on that. also 292 the

line 312 - "There was a dramatic difference" is better

line 313 - "This is may be due to"  better to say This may be due to the fact

line 33 - "it could also be found"  why not simply 'it was found"  ?

line 366 - "Results of this paper were basically" need to match plural subject with plural form of verb

line 370-371 - can you find a clearer way to state this recommendation

Good, well-done science experiment just needs moderate changes to your English presentation to help the reader comfortably flow through the article

Author Response

Response to Reviewer 2 Comments

Point 1: Good, well-done science experiment just needs moderate changes to your English presentation to help the reader comfortably flow through the article. Will need moderate but essential changes to English language issue.

Response 1: Special thanks to you for the affirmation of our scientific research work and paper. We have carefully revised the manuscript based on your comments and recommendations. This manuscript has carefully undergone English language editing by MDPI, which are highlighted in blue colour in the manuscript. The text has been checked for correct use of grammar and common technical terms.

Points 2: line #15 not necessary to capitalize tree name Poplar - lower case as used in line 101 better.  If you agree needs to be changed throughout article.

Response 2: This is a good recommendation, and the capital of poplar was changed into a lower case throughout the whole manuscript.

Point 3: line 34 - confusing last few words "and in future" ?

Response 3: “and in the future” has been changed into a correct expression “in the future”.

Point 4: line 39 -  "forests are expected to"  , drop "greatly".

Response 4: “greatly” has been deleted in the manuscript.

Point 5: line 41 - ", thus yielding lumber, "

Response 5: “thus” has been added into the article.

Point 6: line 46 - "many studies showing that it is" use the gerund form.

Response 6: Corrected, the gerund form was used for “show” in the manuscript.

Point 7: line 51 - "it was worth developing"  delete infinitive "of".

Response 7: “of” has been deleted in manuscript.

Point 8: line 52 - use italics for P. radiata as ou hae done elsewhere in the article.

Response 8: “P. radiata” has been changed into “P. radiata” in the article.

Point 9: line 75 - "more efforts are still needed to find a proper" delete "to put into".

Response 9: “to put into” has been deleted in the article.

Point 10: line 85, "As previously mentioned" also use italics for P radiata.

Response 10: Corrected, italics was used for P. radiata in the article.

Point 11: line 108 "fell down"  ???  do you mean cut down? note lines 123, 124.

Response 11: Yes, “fell down” has been replaced with “cut down” in the manuscript.

Point 12: line 114 - simply say kept and delete "them".

Response 12: “them” has been deleted.

Point 13: line 130 - "like as" is incorrect why not just say either as or like?

Response 13: This is a good recommendation. “like as” has been changed into “like” in the article.

Point 14: line 142, 150 - unless the journal has different rules, one usually italicizes words like "in situ".

Response 14: Thank you so much, this is a really good recommendation. “in situ” has been corrected in italic form “in situ” in this manuscript.

Point 15: line 211 - "A water immersion"  - articles such as  "a" or "the" are missing from certain nouns throughout. Spell check should help on that. also 292 the.

Response 15: Thanks for your kind advices. “A” has been added in line 211 and “The” has been added in line 291.

Point 16: line 312 - "There was a dramatic difference" is better.

Response 16: Corrected. “dramatically” has been changed into “dramatic”.

Point 17: line 313 - "This is may be due to"  better to say This may be due to the fact.

Response 17: “This may be due to the fact” has been changed to “This is may be due to”.

Point 18: line 332 - "it could also be found"  why not simply 'it was found"  ?.

Response 18: Corrected. This sentence has been changed to “it was found”

Point 19: line 366 - "Results of this paper were basically" need to match plural subject with plural form of verb.

Response 19: “was” has been changed to “were” in order to match plural subject with plural form of verb.

Point 20: line 370-371 - can you find a clearer way to state this recommendation.

Response 20: The recommendation in line 370-371 has been modified in a clearer way.

Reviewer 3 Report

The manuscript presents interesting research on the use of ultrasound in seedlings. While the introduction and methods are okay in terms of content, the presentation of results could be improved further by adding more information. Discussion should be improved, especially as to what are the most likely causes of the differences found or what mechanisms are causing the differences. The manuscript is also lacking recommendations for future research, what needs to be studied further and why.

Some minor comments below.

Title: non-specific, I would advise to change it. Application of ... to Poplar Seedling (in order to do what?). Maybe something along the lines of The use of ultrasound on Poplar seedlings or something like that.

17-21: you did not examine all factors equally - MFA was only examined in two trees. Rephrase the sentence.

21-22: sentence starting with Results of this study... This is part of methodology on determining the optimal measuring distance, consider moving it.

26-27: sentence starting with And there ... should be rephrased.

27-29: sentence starting with However: while this is generally true and was confirmed by several studies in the past, your sample size of two trees when examining MFA is in my opinion not enough to talk about definite relationships in the context of this study. Consider rephrasing the sentence.

33-34: it can be measured, but why measure ultrasound TOF if it correlates well with growth days - why not use that instead?

43: lumber is American English for timber (British), pick one and be consistent

47-55: what about juvenile wood, MFA is generally different between juvenile and mature wood. All of the studies cited look at juvenile wood, add more references about juvenile wood and how MFA changes with age.

64: see comment on line 43

64-66: why are there so few studies? Is it due to lack of interest, methods, usability? Expand this paragraph.

80-82: what about the effect of temperature or moisture content? Maybe mention them in recommendations for future research.

108: "from one green" ... Do you mean "from each green"

109-110: explain the basis for this formula, why 15 * root-collar diameter? Why not 12 or 18? Add a reference or justify in text.

123: "prior to being felled" instead of "prior to fell down"
123: add city and country of the manufacturer of the device (Sopron, Hungary), add the same for other devices used in this study. At what frequency do the ultrasonic transducers operate? Add in text.

162-163: why those distances, any particular reason?

168-170: why "proper" and not "optimal" measuring distance? Clearly define what makes it optimal, what criteria were used?

176-178: which year was this measured in, 2018? Add into text.

205: green density of each specimen or overall average density determined in 2.2.3? If second, give the number!

229-230: presumably there were differences in density between top and bottom specimens?

Section 3.1: using the expression "proper" would imply that other distances are improper. Why not use "optimal measuring distance"?

244-245: the statement beginning with However... is only partly true. According to Figure 6, the ultrasonic velocity is increasing in P005 and P006 above 265 mm distance.

Figure 6: x-axis, correct "distence" -> "distance"

Table 1: Add standard deviation or coefficient of variation to represent variability of both root-collar diameter and ultrasound velocity, maybe in brackets beside average values? Why are there 5 seedlings below 10 mm in Sep 2, Sep 20, Nov 5. Did those seedling not grow in diameter, was something wrong with them?

274: maximum average velocity is unclear, rephrase

283-284: correlation does not imply causation! Could be due to longer growth rates, could it also be due to changes in wood density throughout the growing season or something else? Rephrase.

Figure 7: correct x-axis on the left, misspelled.

Section 3.3.1: ultrasonic velocity is likely changing due to the underlying changes in density or other wood properties and not directly due to more growing days. Discuss underlying factors more, as they are likely to have a role in development of mechanical properties that affect the ultrasonic velocity.

Table 2: add number of trees measured (N) on each measuring day into the table

315-316 and Figure 8: will this observed relationship continue with more growth days? Why or why not, discuss.

320: the definition of "root-collar diameter" belongs into Methods, not into Results. Think about moving it.

331: "may do not increase" - rephrase

Figure 9: why not examine average or max/min values as you did in Figure 8?

348-350: rephrase the sentence for clarity starting with However,...

349: rephrase the sentence for clarity starting with Therefore,...

Section 3.3.4: MFA only measured in two trees, why is this not mentioned as a potential problem?

388-391: does MFA change with height, did you account for that?

401: ... are consistent with their reported results.

407-408: does variability (SD or coefficient of variation) change with the method used for calculation of dynamic MOE? Add more information.

427-428: good quality young seedlings are not guaranteeing good quality standing trees. They are a good start, but they are not everything.

Figure 12 and 13: not really surprising, as one Figure is based on the other. Which density was used for calculations of the dynamic MOE?

446-448: I think you are overstating your results here. MFA is important and does affect ultrasound velocity, but your sample size of two trees only confirms the general direction of past findings and is as such not enough to talk about confirmed relationships in the context of this study.

Author Response

Response to Reviewer 3 Comments

Point 1: The manuscript presents interesting research on the use of ultrasound in seedlings. While the introduction and methods are okay in terms of content, the presentation of results could be improved further by adding more information. Discussion should be improved, especially as to what are the most likely causes of the differences found or what mechanisms are causing the differences. The manuscript is also lacking recommendations for future research, what needs to be studied further and why.

Response 1: Special thanks to you for the affirmation of our scientific research work and paper. This article has been carefully revised according to your comments. This manuscript has carefully undergone English language editing by MDPI, which are highlighted in blue colour in the manuscript. The text has been checked for correct use of grammar and common technical terms.

Points 2: Title: non-specific, I would advise to change it. Application of ... to Poplar Seedling (in order to do what?). Maybe something along the lines of The use of ultrasound on Poplar seedlings or something like that.

Response 2: Thanks for your recommendations. The title of this article has been changed to be more specific.

Point 3: 17-21: you did not examine all factors equally - MFA was only examined in two trees. Rephrase the sentence.

Response 3: The sentence in Line 17-21 has been rephrased in the text.

Point 4: 21-22: sentence starting with Results of this study... This is part of methodology on determining the optimal measuring distance, consider moving it.

Response 4: The sentence starting with “Results of this study....” in Line 21-22 has been moved and revised in the text.

Point 5: 26-27: sentence starting with And there ... should be rephrased.

Response 5: The sentence starting with “And there ...” in Line 26-27 has been rephrased in the text.

Point 6: 27-29: sentence starting with However: while this is generally true and was confirmed by several studies in the past, your sample size of two trees when examining MFA is in my opinion not enough to talk about definite relationships in the context of this study. Consider rephrasing the sentence.

Response 6: As reviewer mentioned, the sample size of two trees used for examining MFA is not enough to obtain the definite relationships in this article. Therefore, the sentence in Line 27-29 has been revised.

Point 7: 33-34: it can be measured, but why measure ultrasound TOF if it correlates well with growth days - why not use that instead?

Response 7: Thanks to your comments. Although ultrasonic velocity seems to have a good correlation with growth day within 209 growing days in this article, the relationship between them still need to be verified for longer days. It is more reasonable to measure the ultrasonic speed to evaluate the mechanical properties of seedlings rather than growth days.

Point 8: 43: lumber is American English for timber (British), pick one and be consistent.

Response 8: The term of “timber” has been replaced with “lumber” in the whole text.

Point 9: 47-55: what about juvenile wood, MFA is generally different between juvenile and mature wood. All of the studies cited look at juvenile wood, add more references about juvenile wood and how MFA changes with age.

Response 9: More references about juvenile wood and how MFA changes with age have been added in the manuscript.

Point 11: 64-66: why are there so few studies? Is it due to lack of interest, methods, usability? Expand this paragraph.

Response 11: The paragraph has been expanded in the manuscript.

Point 12: 80-82: what about the effect of temperature or moisture content? Maybe mention them in recommendations for future research.

Response 12: The temperature or moisture content should have an impact on acoustic velocity including ultrasonic velocity, therefore, it is necessary to investigate the effect of temperature or moisture content on ultrasonic speed in future research.

Point 13: 108: "from one green" ... Do you mean "from each green".

Response 13: Yes, “one” has been revised into “each” in the text.

Point 14: 109-110: explain the basis for this formula, why 15 * root-collar diameter? Why not 12 or 18? Add a reference or justify in text.

Response 14: Thanks for your recommendation. We have found that the optimal length for specimen to perform acoustic resonance tests, therefore, the ratio of length to diameter of specimen for acoustic resonance tests was 15 in this paper. The explain for the basis of this formula has been added in text.

Point 15: 123: "prior to being felled" instead of "prior to fell down".

Response 15: Corrected.

Point 16: 123: add city and country of the manufacturer of the device (Sopron, Hungary), add the same for other devices used in this study. At what frequency do the ultrasonic transducers operate? Add in text.

Response 16: The city and country of the manufacturer of the device used in this article has been added in text. And the frequency of ultrasonic transducers was 90kHz, which has already been added in text.

Point 17: 162-163: why those distances, any particular reason?

Response 17: Those distances were chosen in part on the basis of the precision of ultrasonic timer and the convenience of experiment operator. On the other hand, in order to find a proper distance for ultrasonic tests, the wide range of testing distance was required, therefore, a series of values, i.e. 65 mm, 165 mm, 265 mm, 365 mm, 465 mm, 565 mm, 665 mm, 765 mm, 865 mm, and 965 mm were designed.

Point 18: 168-170: why "proper" and not "optimal" measuring distance? Clearly define what makes it optimal, what criteria were used?

Response 18: “Proper” should be used in the line 168-170. From the results of Figure 6, ultrasonic velocity in poplar seedlings kept stable when the detection distance varied from 265mm-665mm, therefore, the proper detection distance for ultrasonic test should be chosen from 265 mm to 665 mm. Considering the convenience and feasibility of experimental tests, the detection distance used in this paper was ultimately set to 365 mm. There is no optimal measuring distance for ultrasonic tests.

Point 19: 176-178: which year was this measured in, 2018? Add into text.

Response 19: The year of measurement was 2018 and this information has been added in the text.

Point 20: 205: green density of each specimen or overall average density determined in 2.2.3? If second, give the number!

Response 20: The density determined in 2.2.3 was green density of each specimen and detailed description has been added in text.

Point 21: 229-230: presumably there were differences in density between top and bottom specimens?

Response 21: The specimens for density determination were cut from the top and bottom of l(mm)-long seedling specimens in order to represent the density of whole seedlings, therefore, the density would be more accurate from the average values of top and bottom specimens. The density between top and bottom specimens may be different due to the different sample position even from the same seedlings.

Point 22: Section 3.1: using the expression "proper" would imply that other distances are improper. Why not use "optimal measuring distance"?

Response 22: As can be seen in Figure 6, ultrasonic velocities of six seedlings all significantly increased when the detection distance changed from 65 mm to 265 mm. And ultrasonic speed in part of seedlings decreased as detection distance was over 665mm. Ultrasonic velocity in poplar seedlings only kept stable when the detection distance varied from 265mm-665mm, therefore, the proper detection distance for ultrasonic test should be chosen from 265 mm to 665 mm. There is no optimal measuring distance for ultrasonic tests.

Point 23: 244-245: the statement beginning with However... is only partly true. According to Figure 6, the ultrasonic velocity is increasing in P005 and P006 above 265 mm distance.

Response 23: Thanks for your recommendations. The statement about Figure 6 in Line 244-245 has been revised in the manuscript.

Point 24: Figure 6: x-axis, correct "distence" -> "distance".

Response 24: Corrected.

Point 25: Table 1: Add standard deviation or coefficient of variation to represent variability of both root-collar diameter and ultrasound velocity, maybe in brackets beside average values? Why are there 5 seedlings below 10 mm in Sep 2, Sep 20, Nov 5. Did those seedling not grow in diameter, was something wrong with them?

Response 25: The standard deviations both for root-collar diameter and ultrasound velocity have been added in the Table 1 shown in brackets besides average values. It should be noted that 60 poplar seedlings were cut down to conduct acoustic resonance tests after the first in situ ultrasonic tests in Jul.24. Therefore, 200 seedlings including 60 new poplar seedlings randomly chosen from nursery were used to conduct second in situ ultrasonic tests in Sep 2. Therefore, there are 5 seedlings below 10mm. Due to the short test duration and low grow rates, there are still 5 seedlings below 10mm in Sep 20. Moreover, because of the bad weather condition, some of poplar seedlings were dead during Sep 20 to Nov.5, therefore, new seedlings with a root-collar diameter lower than 10mm were chosen to complete fourth ultrasonic tests.

Point 26: 274: maximum average velocity is unclear, rephrase.

Response 26: This sentence has been rephrased in the text.

Point 27: 283-284: correlation does not imply causation! Could be due to longer growth rates, could it also be due to changes in wood density throughout the growing season or something else? Rephrase.

Response 27: Thanks to your comments. The sentence in Line 283-284 has been rephrased.

Point 28: Figure 7: correct x-axis on the left, misspelled.

Response 28: Corrected.

Point 29: Section 3.3.1: ultrasonic velocity is likely changing due to the underlying changes in density or other wood properties and not directly due to more growing days. Discuss underlying factors more, as they are likely to have a role in development of mechanical properties that affect the ultrasonic velocity.

Response 29: The underlying factors which are likely to have a role in development of mechanical properties that affect the ultrasonic velocity has been discussed in the manuscript.

Point 30: Table 2: add number of trees measured (N) on each measuring day into the table.

Response 30: The number of trees measured on each measuring day has been added into the Table 2.

Point 31: 315-316 and Figure 8: will this observed relationship continue with more growth days? Why or why not, discuss.

Response 31: The relationship shown in Figure 8 will continue with more growth days. Detailed explanation has been discussed in the text.

Point 32: 320: the definition of "root-collar diameter" belongs into Methods, not into Results. Think about moving it.

Response 32: the definition of "root-collar diameter" has been moved.

Point 33: 331: "may do not increase" - rephrase.

Response 33: The sentence in Line 331 has been rephrased in text.

Point 34: Figure 9: why not examine average or max/min values as you did in Figure 8?

Response 34: It is meaningless to calculate the average or max/min values for Figure 9 as did in Figure 8. The reason why we calculate the average values for velocity in Figure 8 is that there are 145 ultrasonic velocity data for one test date, such as 105 days later. Therefore, we need to calculate the average velocity for each test date to analyse the relationship between growth day and ultrasonic velocity. However, each datapoint in Figure 9 has a specific root-collar diameter and corresponding ultrasonic velocity. Each point represents one seedling sample, therefore, it is not accurate and reasonable to calculate the average values when analyse the relationship between ultrasonic velocity and root-collar diameter of seedlings.

Point 35: 348-350: rephrase the sentence for clarity starting with However,...

Response 35: This sentence has been rephrased in the manuscript.

Point 36: 349: rephrase the sentence for clarity starting with Therefore,...

Response 36: This sentence has been rephrased in the text.

Point 37: Section 3.3.4: MFA only measured in two trees, why is this not mentioned as a potential problem?

Response 37: This is a really useful comment. The number of sample used for MFA was not enough in this paper, therefore, this should be a potential problem when talking about the effect of MFA on ultrasonic velocity in seedlings. These are in detail described in the text.

Point 38: 388-391: does MFA change with height, did you account for that?

Response 38: The relationship between height and MFA was not investigated in this paper, therefore we did not account for that. It is necessary to investigate the relationship between MFA and the height of seedling, and we will do it in the further research.

Point 39: 401: ... are consistent with their reported results.

Response 39: Corrected.

Point 40: 407-408: does variability (SD or coefficient of variation) change with the method used for calculation of dynamic MOE? Add more information.

Response 40: The standard deviations (SD) of ultrasonic velocity and acoustic velocity as well as their corresponding dynamic MOE were added in the text.

Point 41: 427-428: good quality young seedlings are not guaranteeing good quality standing trees. They are a good start, but they are not everything.

Response 41: Thanks to your comments. As reviewer mentioned, even though good quality young seedlings are not guaranteeing good quality standing trees, but they are a good start to cultivate the high properties of plantation trees.

Point 42: Figure 12 and 13: not really surprising, as one Figure is based on the other. Which density was used for calculations of the dynamic MOE?

Response 42: Green density was used for calculations of the dynamic MOE.

Point 43: 446-448: I think you are overstating your results here. MFA is important and does affect ultrasound velocity, but your sample size of two trees only confirms the general direction of past findings and is as such not enough to talk about confirmed relationships in the context of this study.

Response 43: This is a really good recommendation. The statement of results in Line 446-448 has been revised in the text.

This manuscript is a resubmission of an earlier submission. The following is a list of the peer review reports and author responses from that submission.

Round 1

Reviewer 1 Report

I have revised the paper “Ultrasonic method for evaluationg wood quality of poplar seedlings”.

The paper is well written and clear and the English is good. The methods and sampling are correct.

On the contrary the general scientific content of the paper is poor and the authors should work more to give a scientific answer to their question. If the question is “Does the quality of the seedling wood can be assessed by acoustic and ultrasound measurements?”, the question is without answer.

Authors observe that the ultrasound speed is well related to plan age but they do not ask themselves why.

No answer about it are provided. If the ultrasound speed depends mainly on the age of the plant is clear how this technique is not able to assess wood quality….

Talking about density is not mentioned how the density was determined. The relation between ultrasonic speed and density have not a great interest and the relation is poor because ultrasound speed, as well known does not depends only on density.

Any objective method to determine the quality of wood was used. I mean that dynamic MOE shoul be plot against some objective wood quality indexes such as real bending or tension tests or grain angle compared to stem axis. I mean that indirect quality measurement should be plot against some direct measurement of wood quality otherwise the paper has no sense.

At present the paper is a well done exercise with lack of scientific meaning.

Reviewer 2 Report

I do have concerns with the relationship between growth days and ultrasonic velocit because I believe there are no significant growth days under study to take conslusions. Also it is not clear the interest and relevance of using US for predict wood quality and there are no bibliographic support to acknowledge the relevance of the results.